# Health Benefits of Antioxidant Bioactive Compounds in the Fruits and Leaves of *Lonicera caerulea* L. and *Aronia melanocarpa* (Michx.) Elliot

**DOI:** 10.3390/antiox12040951

**Published:** 2023-04-18

**Authors:** Bogdan-Stefan Negreanu-Pirjol, Ovidiu Cristian Oprea, Ticuta Negreanu-Pirjol, Florentina Nicoleta Roncea, Ana-Maria Prelipcean, Oana Craciunescu, Andreea Iosageanu, Victoria Artem, Aurora Ranca, Ludmila Motelica, Anca-Cristina Lepadatu, Madalina Cosma, Dan Razvan Popoviciu

**Affiliations:** 1Faculty of Pharmacy, Ovidius University of Constanta, Capitan Aviator Al. Serbanescu Street no. 6, Campus, Corp C, 900470 Constanta, Romania; 2Department of Inorganic Chemistry, Physical Chemistry and Electrochemistry, Faculty of Chemical Engineering and Biotechnologies, University Politehnica of Bucharest, Gh. Polizu no. 1-7, 011061 Bucharest, Romania; 3National Research Center for Food Safety, University Politehnica of Bucharest, Splaiul Independentei no. 313, 060042 Bucharest, Romania; 4National Center for Micro and Nanomaterials, University Politehnica of Bucharest, Splaiul Independentei no. 313, 060042 Bucharest, Romania; 5Academy of Romanian Scientists, Ilfov Street 3, 050044 Bucharest, Romania; 6National Institute of R&D for Biological Sciences, Splaiul Independentei no. 296, 060031 Bucharest, Romaniaandreea.iosageanu@incdsb.ro (A.I.); 7Research-Development Station for Viticulture and Winemaking of Murfatlar, Calea Bucuresti no. 2, Constanta County, 905100 Murfatlar, Romania; 8Faculty of Natural Sciences and Agricultural Sciences, Ovidius University of Constanta, University Alley no.1, Campus, Corp B, 900470 Constanta, Romania

**Keywords:** haskap, *Aronia*, polyphenols, anthocyanins, flavonoids, iridoids, flavonoids, antioxidant, anti-inflammatory, antimicrobial, antitumor, hepatoprotective, encapsulation

## Abstract

*Lonicera caerulaea* L. and *Aronia melanocarpa* (Michx.) Elliot fruits are frequently used for their health benefits as they are rich in bioactive compounds. They are recognized as a source of natural and valuable phytonutrients, which makes them a superfood. *L. caerulea* presents antioxidant activity three to five times higher than other berries which are more commonly consumed, such as blackberries or strawberries. In addition, their ascorbic acid level is the highest among fruits. The species *A. melanocarpa* is considered one of the richest known sources of antioxidants, surpassing currants, cranberries, blueberries, elderberries, and gooseberries, and contains one of the highest amounts of sorbitol. The non-edible leaves of genus *Aronia* became more extensively analyzed as a byproduct or waste material due to their high polyphenol, flavonoid, and phenolic acid content, along with a small amount of anthocyanins, which are used as ingredients in nutraceuticals, herbal teas, bio-cosmetics, cosmeceuticals, food and by the pharmaceutical industry. These plants are a rich source of vitamins, tocopherols, folic acid, and carotenoids. However, they remain outside of mainstream fruit consumption, being well known only to a small audience. This review aims to shed light on *L. caerulaea* and *A. melanocarpa* and their bioactive compounds as healthy superfoods with antioxidant, anti-inflammatory, antitumor, antimicrobial, and anti-diabetic effects, and hepato-, cardio-, and neuro-protective potential. In this view, we hope to promote their cultivation and processing, increase their commercial availability, and also highlight the ability of these species to be used as potential nutraceutical sources, helpful for human health.

## 1. Introduction

Reactive oxygen species (ROS) represent a group of molecular oxygen derivatives that are formed as natural by-products of oxygen metabolism. Chemically, ROS are activated forms of oxygen that contain one or more unpaired electrons and can be either free radicals, such as superoxide radicals (O^2−^), hydroxyl radicals (-OH), and hydroperoxyl radicals (-HO_2_), or non-radical species, such as hydrogen peroxide (H_2_O_2_), singlet oxygen (^1^O_2_), and ozone (O_3_) [1].

Experimental evidence suggests that oxidative stress plays an important role in the progression of more than 100 diseases, including cancer, cardiovascular diseases, and neurodegenerative disorders [2]. The damage caused by oxidative stress can be avoided by endogenous and exogenous antioxidant systems. The endogenous antioxidant defense mechanism is referring to enzymes, such as superoxide dismutase, catalase, and glutathione, while the exogenous antioxidant defense mechanism is referring to dietary compounds, especially compounds of plant origin, such as phenolic acids, flavonoids, and vitamins [3,4,5,6,7].

It has been suggested that the antioxidants of plant origin possess antimicrobial, antiviral, anti-inflammatory, anti-hypertensive, hepatoprotective, neuroprotective, antidiabetic, and antitumor properties [8]. Because of this, many plants are traditionally used as folk medicines by millions of people around the world [9,10].

Today, the health of the population is affected by environmental pollution and poor nutrition. The high incidence of several chronic diseases has also increased the demand for superfoods with health benefits [11]. Some epidemiological and interventional studies have demonstrated a positive correlation between the consumption of polyphenolic-rich fruits and reduced chronic diseases [12]. Medicinal plants have been used since ancient times to treat or prevent various diseases, and researchers are now observing them with increased attention [4,13,14,15].

In Romania, forest fruit production is continuously growing due to the valuable pedo-climatic conditions, easy implementation of food safety [16], potential for significant economic benefits and exports [17], and a longer harvesting period than other Central European countries. As a result, *A. melanocarpa* and *L. caerulea* were selected for the present review in order to provide scientific data on their potential valorization due to their bioactive compound content and biological properties.

In this context, the present review aims to present recent research on *Lonicera caerulaea* L. and *Aronia melanocarpa* (Michx.) Elliott species and their bioactive compounds, mainly phenolic compounds, as superfoods with valuable biological properties [18], such as antioxidant, anti-inflammatory, antimicrobial, and anti-diabetic effects, and hepato-, cardio- and neuro-protective activity, which makes them beneficial for human health. In this view, we target promoting their cultivation, processing, and increased commercial availability, besides highlighting the usefulness of these species as important ingredients of nutraceuticals for human health.

## 2. *Lonicera caerulea* Plant

The *L. caerulea* berry is an edible blue fruit native to the boreal forests and low-lying wet regions of northern hemisphere countries, including Canada, Japan, and Russia [19]. It is also known by its common names, which include haskap, blue honeysuckle [20] sweet-berry honeysuckle [21], fly honeysuckle [22], and blue fly honeysuckle, and is a member of the Caprifoliaceae family, closely related to decorative honeysuckles in the same genus. Haskap is the name used by the native Ainu people of the northern islands of Japan, where other subspecies also grow. Their word for them, “Haskaps”, translates to “Edible Blue Honeysuckle”.

It is a circumpolar species, with various local varieties in Northern and Central Asia, Eastern Europe, and North America, and is considered a fruit and ornamental shrub.

*L. caerulea* var., native to Eastern Siberia and Northern Japan, is mostly cultivated throughout the world for its applications in folk medicine, or as we said, as a “superfood”. Its applications in traditional medicine have been documented for hundreds of years, being used for its antipyretic and anti-inflammatory properties, as well as in the treatment of digestive, ophthalmological, and cardiovascular disorders [23,24]. Numerous research studies have been conducted in order to describe the phytoconstituents of *L. caerulea* and, to a certain extent, their influence on human health has also been described [25].

*L. caerulea* is a frost-resistant plant that can withstand temperatures down to −40 °C in winter and up to 35 °C in summer. However, the temperature increase might affect the vegetative growth of the plant, compared to other species of shrubs. The plant is a light-loving species, preferring sunny places, but the limiting thermal factor is represented by the return of frost because *L. caerulea* blooms in early spring as the first fruiting shrub, and its flowers are frequently affected. At the first positive temperatures in spring, *L. caerulea* plants start growing, blooming, and fruiting after the 15th of March, and any return of frost in the March–May period will totally compromise the harvest by seriously affecting the vegetative growth. The plant prefers fertile soils, rich in organic matter (sandy–loamy, loamy), and well-drained and regenerated [26].

It was first introduced in Romania in 1988 [27,28] and is cultivated in gardens and commercial plantations, with varieties represented by Nymfa, Loni, Cera, and Kami.

The haskap fruits ripen early (at the end of April–beginning of May), and have an ovoid or obovate shape. They are convex in the middle, dark blue in color, covered with pith, weigh 0.7–1.3 g depending on the variety, and are up to 2 cm long and 0.5 cm in diameter. The fruits have food value and are consumed fresh or processed in the form of jams, jellies, soft drinks, and candied infusions, as well as wine through fermentation [29].

## 3. *Aronia melanocarpa* Plant

The *Aronia* species belongs to the Rosaceae family, subfamily Amygdaloideae, tribe Maleae, subtribe Malinae, commonly known as chokeberry, and comprises three species: *A. arbutifolia* (L.) Pers. (red chokeberry), *A. melanocarpa* (Michx.) Elliot (black chokeberry), and *A. prunifolia* (Marshall) Rehder (purple chokeberry). It originated in North America and then spread all over the world, mainly in the 20^th^ century in Central and Southeastern Europe (Croatia, Serbia, Bulgaria, Romania, Poland, Germany), and Southeastern Asia (Russia, Korea) [30,31].

*Aronia* sp. is a frost-resistant shrub that can withstand winter temperatures as low as −30–35 °C. It is a light-loving species (it prefers sunny places), but it also tolerates partial shade. For plants in semi-shade, production will be lower and of lower quality.

*Aronia* plants do not have special needs in terms of climatic and eco-pedological factors, nor do they register strong disease and pest attacks. They tolerate drought but for short periods. In conditions of excessive drought, the fruits lose their juiciness/turgidity, harden, and raisin. *Aronia* does not tolerate excessive soil moisture, but drip irrigation is mandatory in the hot and dry periods of summer. The plant prefers well-drained and regenerated medium (sandy-loamy, loamy) and light (sandy) soils. However, it tolerates heavier soils (clay) but does not support calcareous soils (with carbonates), in which it shows pronounced chlorosis. In Romania, the varieties of chokeberry that are planted are *A. melanocarpa* “Nero” and *A. melanocarpa* “Melrom” [32].

*Aronia* fruits (*A. melanocarpa*) are edible drupes, juicy, slightly sweet, with an astringent taste. The color of the skin is glossy-black, and the flesh is dark red. Flowers are around 5 cm in diameter, white, with five petals each. Fruits are small (around 1 cm in diameter), spherical, with purple-black pomes, forming in mid to late summer. Each fruit contains 1–5 seeds. Besides their valuable composition, rich in bioactive compounds with important biological properties, these fruits could be valorized at an industrial scale by easy processing into edible jams and as stable food colorants [33].

They are harvested at the end of July–August when they have softened a little and have a slightly sweet taste. A delay in harvesting leads to raisining and loss of juiciness, and they become unfit for fresh or prepared consumption. The fruits have food value and could be consumed fresh or preserved in preparations. While still underused, *A. melanocarpa* fruits are considered rather tasty (mostly when processed, rather than raw) due to their tannin content as juice, syrup, jelly, and jam, and can be used for flavoring food products such as tea, ice cream, and yogurt. Recently, *A. melanocarpa* seeds were used as raw materials for the extraction of high-quality oil by Piasecka et al. [34,35]. Chokeberry oil was found to have the highest linoleic acid content among all studied oils (raspberry, blackberry, and chokeberry), while the major tocols fraction was α-tocopherol [36].

Some recent research indicates the possible use of chokeberry pomace extract as an additive for apple juice, helping with its preservation and positively affecting the color and taste due to its antioxidant colorants [35,37,38]. The high anthocyanin content made chokeberry pomace suitable as an ingredient for chitosan-based packaging films with capacity to indicate pH variations [39]. At the same time, enriching rape honey with chokeberry fruit additive increases the antioxidant activity three to 15 times, and intensifies its antibacterial and antiviral activities [40]. Similar strong activities were also reported by [41], which recommends the use of chokeberry and honeysuckle ethanol infusion as a food additive, especially against *Listeria monocytogenes*.

## 4. Bioactive Compounds of *Lonicera caerulea* Fruits and Leaves

The search for nutritionally and biologically superior superfoods continues to grow and an emerging plant that fulfills this quest is *L. caerulea.* One of the most notable characteristics of *L. caerulea* fruits is their antioxidant effect, owing to their high vitamin C and phenolic content. It has been shown that *L. caerulea* presents antioxidant activity three to five times higher than some more commonly consumed berries, such as blackberries or strawberries [42]. The ascorbic acid content varies between 30.5 to 186.6 mg·100 g^−1^, with an average of 53.5 mg·100 g^−1^, meaning this fruit is regarded as a “superfood” [43].

### 4.1. Composition in Bioactive Compounds

Haskap fruits are a rich source of nutrients and bioactive compounds. They represent a valuable source of natural and valuable phytonutrients, such as polyphenols, flavonoids, anthocyanins, minerals, and secondary metabolites with bioactive properties that are important for maintaining proper human health [21]. Total sugars average around 7.20% of the fruit biomass, the dominant being glucose and fructose (each with around 3%), followed by bound saccharides. Lipids constitute only 1.50%, of which around 0.90% are fatty acids, especially palmitic and oleic acids, followed by stearic, myristic, linoleic, palmitoleic, and lauric acids. Sterols, hydrocarbons, and alcohols are also present in small amounts. Around 15% of fruit mass is fiber, while organic acids compose 12% (mostly citric and malic) [27].

### 4.2. Phenolic Compounds

Nevertheless, the most important component of haskap fruit is the phenolic fraction, with around 4% of the fresh biomass. Polyphenols are a very large group of compounds, with different structures having physical, chemical, and biological properties. Depending on the position of the phenolic ring and the degree of oxidation of the pyranone ring, we distinguish the following eight groups: phenolic acids (hydroxycinnamic acids, hydroxybenzoic acids), isoflavones, flavones, flavonols, flavanones, flavan-3-ols, anthocyanins, and chalcones [44,45,46,47]. These compounds are considered very important because they have the ability to neutralize free radical activity [48].

#### 4.2.1. Phenolic Acids and Flavonoids

The phenolic compounds found in *L. caerulea* fruits include phenolic acids (caffeic, chlorogenic, and neochlorogenic acids), catechins, and flavonoids (quercetin) [49]. However, phenolic acid content has been found to vary in accordance with the cultivar, growing conditions, and time of harvest [50]. Phenolic acids range between 2800–5500 mg/kg DW, and chlorogenic, caffeic, and ferulic acids are dominant, although coumaric, vanillic, salicylic, and others also occur in small concentrations. Among flavonoids, quercetin, along with quercetin glucosides and rutinosides, are the most abundant (known for their antitumoral properties). Apigenin (antitumoral and anti-inflammatory), catechin, and epicatechin also form a significant fraction. Regarding the phenolic acids, they are divided into hydroxybenzoic acids with carbon skeleton C6–C1, derived directly from benzoic acid and hydroxycinnamic acids built on skeleton C6–C3 [51] (Figure 1).

Zadernowski et al. [53] showed that 61.1% of the total phenolic acids of haskap fruits consisted of hydroxycinnamic acids (mostly chlorogenic acid) and their derivatives (p-coumaric acid and m-coumaric acid).

Jurikova et al. [45] identified chlorogenic acid as the main phenolic acid in the range of 86.62–267.14 mg/100 g fresh weight (FW), while Svarcova et al. [27] found that the total content of phenolic acids in *L. caerulea* fruits ranged from 2845.8 ± 141.0 to 5418.2 ± 228.0 mg·kg^−1^, dry weight (DW), and chlorogenic acid content was 0.42%. Kucharska et al. [54] reported that the chlorogenic acid content ranged from 17.24–60.37 mg/100 g FW. Also, Senica et al. [44] reported a chlorogenic acid content of 22.45–46.06 mg/100 g FW.

Khattab et al. [55] reported that the chlorogenic acid content ranged from 35.0 to 44.0 mg/100 g FW in some varieties grown in Canada.

Chlorogenic acid (Figure 2) is one of the main polyphenols in the human diet. It possesses many health-promoting properties and has been found to have anti-oxidant [56], anti-inflammatory [56], and anti-cancer [57,58,59,60,61] activities.

Onakpoya et al. and Zhao et al. have proved that chlorogenic acid shows antihypertensive properties [62,63].

#### 4.2.2. Anthocyanins

Anthocyanins are the predominant compounds found in *L. caerulea* fruits. Anthocyanins are responsible for the intense color of the fruits and are also known for their antibacterial activities. Out of this class of flavonoids, cyanidin and its glucosides (cyanidin-3,5-diglucoside) and rutinosides, peonidin and its glucosides (peonidin-3-glucoside), delphinidin, pelargonidin, and other anthocyanins have been recently described [64]. The most abundant is cyanidin-3-glucoside, representing around 79–92% [65]. Several studies on these plants’ secondary metabolites, particularly cyanidin, have proved their antioxidant properties [66].

Thus, the rich content in anthocyanidins differentiates *L. caerulea* from other edible *Lonicera* species. Finally, another group of phenols present in haskap fruits are proanthocyanidins [27,67,68]. Haskap fruits are known for their high content of anthocyanins (from 400 to 1500 mg/100 g) [45]. Gruia et al. [69] reported 8.21 ± 0.32 mg of total anthocyanins per 100 mL aqueous extract.

Anthocyanidin content was also reported in the range between 1500–6500 mg/kg FW [54]. Mladin et al. [32] reported a mean content of 4439 mg of total anthocyanins per kg, with the highest content in the skin of the fruit (4870 mg/kg) and a lower content in the flesh of the fruit (309–791 mg/kg).

Caprioli et al. [70] showed that cyanidin-3-glucoside is the major constituent of anthocyanins (10.763 mg/g DW) representing 87% of the total anthocyanins amount.

Khattab et al. [71] reported that total anthocyanin content, expressed in cyanidin-3-glucoside, was in the range of 4.49 ± 0.14 mg/g FW to 6.97 ± 0.16 mg/g FW.

Cesoniene et al. [72] reported that the anthocyanins comprised 53.8% of total phenolic contents and ranged from 277.8 ± 1.1 mg/100 g to 394.1± 8.4 mg/100 g FW, cyanidin-3-glucoside being the main anthocyanin present in 11 cultivars of *L. caerulea*.

Wang et al. [73], Rupasinghe et al. [42], and Ochmian et al. [50] reported that the major anthocyanin is cyanidin-3-glucoside (79–92%), while cyanidin-3,5-diglucoside, peonidin-3-glucoside, cyanidin-3-rutinoside, peonidin-3-rutinoside, and pelargonidin-3-glucoside appeared in smaller quantities. Wang et al. [73] also found that disaccharide anthocyanins, such as cyanidin-3,5-O-dihexoside and peonidin-3,5-O-dihexoside were present in haskap. For the first time, the presence of acylated anthocyanins, such as cyanidin-3-O-acetylhexoside and peonidin-3-O-acetylhexoside, was described in haskap fruits [73,74].

Liu et al. [74] found that one of the anthocyanins present in the fruits was cyanidin-3-sophorose-5-glucoside, along with cyanidin-3-hexoside-ethyl-catechin [27]. Other anthocyanins also found in fruits were pelargonidin-3-O-glucoside, pelargonidin 3, 5-O-diglucoside [28,31,46,54,67,70,71,75], peonidin-3-O-rutinoside and peonidin-3-O-glucoside [31,46,50,54,70,71,73], delphinidin-3-O-glucoside and delphinidin-3-O-rutinoside [67,73,76,77], 5-methylpyranocyanidin-3-hexoside, and 5-methylpyranopeonidin-3-glucoside [78].

According to Ponder et al. [79], the predominant anthocyanin found in three cultivar haskap fruits, harvested and analyzed during 2018–2019, was cyanidin-3-glucoside. The total anthocyanins content, expressed as cyanidin-3-glucoside, was between 443.04 ± 50.49 and 1376.46 ± 141.60 mg CGE/100 g DW.

The structure of some anthocyanins is presented in Figure 3. The differences regarding anthocyanin content reported in several studies might be attributed to the varietal and genetic features of haskap, as well as to extraction conditions.

Anthocyanins also maintain a high level in fermented products, such as haskap wine. From about 130 mg/L in fresh juice, they may drop to 60 mg/L after fermentation and less (30 mg/L) if wine is subjected to aging [29].

#### 4.2.3. Other Compounds

Iridoids (a group of monoterpenoids) are one of the main bioactive compound classes (especially antitumoral, antibiotic, hepato- and neuroprotective, hypotensive, and anti-inflammatory) found in haskap fruits. This is important since few fruits are known to contain significant amounts. Loganic acid is dominant in haskap, while other iridoids, such as loganin and loganin pentosides, secologanin, secoxyloganin, sweroside, and pentosyl-sweroside, are found in lower concentrations [80]. Iridoids are also abundant in these fruits, with 1200–2700 mg/kg FW [54].

Vitamin C is another component with important health benefits, with concentrations of 85–300 mg/kg, and even up to 1860 mg/kg FW according to some authors, which are among the highest found in known fruits [28,81]. Additionally, vitamin E in haskap fruits may range between 0.90–3.70 mg/kg DW [82].

Among minerals, iron, potassium, and manganese are the most abundant, with 10,000–30,000 mg/kg DW [68].

#### 4.2.4. Bioactive Compounds of *Lonicera caerulea* Leaves

The leaves of haskap were recently found to be an important source of bioactive compounds. According to the variety, the total phenolic content varied between 32.127 and 52.399 mg/g, carotenoids content was between 1.848–2.876 mg/g, while chlorophyll A and chlorophyll B were present in concentrations of 10.642–14.003 and 4.463–5.575 mg/g, before storage [83]. The leaf extract prepared by ultrasonication in distilled water at 30 °C for 60 min contained several phenolic acids, such as loganic acid with anti-inflammatory properties acting against hyaluronidase and hypolipidemic potential acting against lipase, chlorogenic acid with antidiabetic potential acting against α-glucosidase, and caffeic acid with antioxidant and anti-inflammatory activity [83]. An extract prepared by aqueous two-phase flotation using ammonium sulfate-ethanol presented increased yields of flavonoids, such as rutin, luteolin-7-O-glucoside, and diosmin, which provided increased antioxidant capacity [84,85]. Vitamin C was also determined in the aqueous extract of haskap leaves prepared at the Soxhlet equipment operated at 105 °C, for 20 h, but at a lower value (5.1 mg/100 g) than in fruit extract (6.9 mg/100 g) and polysaccharides (7%), together with flavones, flavanones, and tanning compounds, which could provide antibacterial and antiviral activity [86].

## 5. Bioactive Compounds of *Aronia melanocarpa* Fruits and Leaves

*A. melanocarpa* is frequently used for its health benefits due to the richness of the bioactive compounds of its fruits. It is well known that the fruits possess strong free radical scavenging potential due to an increased content of procyanidins, anthocyanins, flavonoids, and phenolic acids [87]. The chemical composition of *A. melanocarpa* L. leaves differs depending on the cultivar, climatic zone, maturity stage, and the extraction method used, as mentioned in numerous articles [18,87,88,89]. Mixed spectrophotometric and chromatographic methods determined the chemical composition of *Aronia* leaves, from total phenolic content expressed as flavonols, flavonoids, anthocyanins, phenolic acids [87,89,90,91,92,93,94,95], small amounts of chlorophyll, carotenoids [94], carbohydrates, macro and microelements, fibers, and crude fat [96], with mostly high levels of chlorogenic acid, caffeoylquinic acid derivatives, quercetin, rutin, sorbitol, and small amounts of anthocyanins, chlorophyll, carotenoids, macro and microelements, and fibers identified mostly in young leaves compared to mature and old leaves (Figure 4).

The extracts obtained from nonedible *A. melanocarpa* L. leaves have potential use as they have antioxidant, healing, hypoglycemic, antineurodegenerative, and moderate antimicrobial activity.

Although a valuable raw material for developing a sustainable economy, there is still little information regarding the leaves of *A. melanocarpa* L., which are considered by some authors to be a byproduct or waste material [89].

The antioxidant potential of *A. melanocarpa* leads to diverse antibacterial, antiviral, anti-hypertensive, anti-aggregating (anti-coagulant), antilipidemic, anti-diabetic, cardio-, gastro- and hepatoprotective, anti-inflammatory, immunomodulatory and radioprotective activity [97].

### 5.1. Bioactive Compounds

Fruits are rich in sugars (12–20% of total soluble solid compounds), and have an acid pH (3.3–3.7) [98,99]. Glucose and fructose are dominant. *Aronia* fruits are known to contain one of the highest amounts of sorbitol (80 g/L in fresh juice), a sweet alcoholic compound used as a substitute for sugar.

Lipids are around 1400 mg/kg FW, mostly found in seeds (over 19,000 mg/kg). Linoleic acid is the main fatty acid. Phospholipids (phosphatidylcholine, phosphatidylinositol, phosphatidylethanolamine) are another important component of chokeberry seeds, along with various sterols. Proteins make up 7000 mg/kg of the fruit’s fresh weight, with asparagine as the main amino acid. Various organic acids are to be found, such as malic, citric, isocitric, shikimic, and succinic acid, with a total of up to 15,000 mg/kg.

Potassium, calcium, and magnesium are the main minerals in chokeberries [100]. Among microelements, iron, manganese, silicon, and boron are abundant (2–20 mg/kg) [88].

Chokeberry is considered one of the richest sources of antioxidants known, surpassing currants, cranberries, blueberries, elderberries, and gooseberries [98]. Another important chemical with antioxidant properties is ascorbic acid (vitamin C). Vitamin C makes up around 120 mg/kg of the fresh biomass. However, vitamin C is known to be highly unstable. Experiments have shown that while fruits can maintain their levels of ascorbic acid over time, degradation occurs rather rapidly in *Aronia* juice, with over 15% in three days. Thus, processed chokeberry products need to be consumed fast [101].

Apart from vitamin C, chokeberries are a rich source of B complex vitamins (B_1_, B_2_, B_3_, B_5_, B_6_), vitamin K, tocopherols, folic acid, and carotenoids. Among carotenoids, the most abundant are β-carotene, β-cryptoxanthin, and violaxanthin, all considered to be potent antioxidants. A specific group of chemicals, depsides (a type of polyphenols with aromatic nuclei joined by ester links) with potential pharmaceutical and cosmetic applications, is also present [18].

Apart from fresh fruits and juice, *Aronia* pomace was found to be rich in key compounds, such as fibers (over 55%), sugars (12–15%), vitamin C (90–260 mg/kg), and minerals (2–3%), making pomace powder a potential functional ingredient in fortifying pastry and bakery products or as a dietary supplement [102].

#### 5.1.1. Phenolic Compounds of Aronia Fruits

One of the most important classes of bioactive compounds isolated from chokeberries is phenolic compounds, substances with high antioxidant potential [103]. *A. melanocarpa* fruits are among the richest in polyphenols. More precisely, they are rich in compounds such as procyanidins, anthocyanidins, and phenolic acids, but have surprisingly low flavonols content. The ripening of the fruits leads to changes in the levels of polyphenol content, and the latest research has demonstrated that the cultivation of *A. melanocarpa* shrubs in a dry and warm climate leads to the concentration of the polyphenol content in the fruit.

Phenolic acids represent 7.5% of the total amount of polyphenolic compounds in *A. melanocarpa* extracts since the main constituent part of phenolic acids is represented by chlorogenic acid and neochlorogenic acid [104]. The major phenolic acid constituent is chlorogenic acid, which in turn is made up of caffeic acid that is linked to quinic acid through an ester bond [105]. In *Aronia* fruits, the mainly accumulated chlorogenic acid is 3-caffeoylquinic acid (~69.4 mol% of the sum of chlorogenic acids), and 5-caffeoylquinic and 4-caffeoylquinic acids (~14 mol%) in smaller but comparable amounts [106]. Both chlorogenic and non-chlorogenic acids are also found in the leaves of *A. melanocarpa*.

Almost 40% of the antioxidant activity in *A. melanocarpa* is given by procyanidins [107]. A decrease in the amount of procyanidins was observed as the fruits started to ripen; more precisely, the amount of procyanidins compounds decreased significantly when the fruits were fully ripened [108].

The total concentration can reach up to 30 g of phenols (of which 3–8 g anthocyanins) per kilogram of fresh fruit biomass. The most abundant phenols are cyanidin-3-glycosides, epicatechin, and proanthocyanidins (condensed tannins), which make up 60% of the total amount [97].

The latest research discovered that the antioxidant activity of anthocyanins is responsible for certain biological activities, such as preventing and reducing the risk of cardiovascular diseases, cancer, and diabetes [109]. These compounds are known to be part of the flavonoid family, but are distinguished by their ability to form flavylium cations [110]. Anthocyanins represent approximately 25% of the polyphenol content found in *A. melanocarpa* fruits. Anthocyanins can also be divided into 4 groups of cyanidin glycosides: 3-O-galactosides (68.9%), 3-O-glucosides (1.3%), 3-O-arabinosides (27.5%) and 3-O-xyloside (2.3%) [104].

The stability of anthocyanins is affected by the environmental conditions, so that a decrease in their quantity can be observed following sudden changes. One physical factor, such as light, could create a self-association effect, while another factor, the thermal treatment, could lead to a rapid and linear decrease in the anthocyanin content [111]. After several tests, it was observed that storing the fruits of *A. melanocarpa* at a temperature of 70 °C for 24 h could lead to a decrease of 50% in the content of anthocyanins [112].

#### 5.1.2. Phenolic Compounds of *Aronia* Leaves

In regard to the non-edible leaves of the *Aronia* genus, they became more extensively analyzed as a by-product or waste material, due to their high polyphenol, flavonoid, and phenolic acid content, along with small amounts of anthocyanins (2 mg/100 g) [91]. The diversity of chemical compounds recommends *Aronia* leaf extracts as valuable ingredients in nutraceuticals and herbal teas and in the biocosmetics, cosmeceuticals, food, and pharmaceutical industries [113].

It is important to notice that the chemical composition of *A. melanocarpa* leaves differs between cultivars, climatic zones, harvesting time, and maturity stage, but also with pretreatment conditions, extraction methods, and conditions, as mentioned in numerous articles [18,87,88,89]. Accordingly, the extracts vary in their in vitro and in vivo antioxidant and pharmacological capacity. Mixed spectrophotometric and chromatographic methods determined the chemical composition of *Aronia* leaves, from total phenolic content expressed as flavonols, flavonoids, anthocyanins, phenolic acids [87,89,90,91,92,93,94,95], small amounts of chlorophyll, carotenoids [94], carbohydrates, macro and microelements, fibers, and crude fat [96], with most of high levels for chlorogenic acid, caffeoylquinic acid derivatives, quercetin, rutin, sorbitol, small amounts of anthocyanins, chlorophyll, carotenoids, macro and microelements, fibers identified mostly in young leaves compared to mature and old leaves.

Various *A. melanocarpa* leaves originated from plantations [90,97], an ecological type, var. ‘Nero” [95], arboretum [91], allotment gardens [96] and Orchards Company Trzebnica, (Michx.) Elliott; cv.: ‘Galicjanka’ [93], local farm var. “Nero” [94], were collected. They were at different developmental stages, from young (2 weeks), mature (2 months), and aged (4 months) [92,94]. They were collected from June to October [95], from July to September [91], and in August [87], from one-year-old shoots without fruit [96], and no data were mentioned [90,93,97].

Prior to extraction, *A. melanocarpa* leaves were submitted to various pretreatments, including inspection for insect and mechanical damage [93,94], and were washed with water [92,93] then dried in different conditions from shade [90], outside in the open air at 25 ± 2 °C for 10 days [91], naturally on draft in the dark until their moisture content reached 10% [87], freeze-dried [94], then ground, sieved [90,94], frozen, lyophilized [93,94,95], lyophilized [92], and finally preserved at (−70 °C, −80 °C until use) in the dark [87].

The widely used extraction methods varied from conventional to environmentally friendly ones [87] using solid/liquid extraction with or without reflux, in the presence of different polar solvents (water, boiled deionized water, ethanol 70% (*v*/*v*) with or without heat, 80% methanol acidulated with 30% HCl, 2% ascorbic acid or acetic acid) [90,91,92,93,94,95,97], maceration in 70% ethanol [89], and subcritical water extraction, because of its safety, green character, and low cost [87], followed by techniques of separation and purification.

Although it is a valuable raw material for developing a sustainable economy there is still little information about *A. melanocarpa*, especially leaves, considered by some authors as a by-product or waste material [89]. In this regard, only the 14th edition of the State Pharmacopoeia of Russia includes monographs for *A. melanocarpae* recens fructus FS.2.5.0002.15 and *A. melanocarpae* siccus fructus FS. 2.5.0003.15 as a tonic and adaptogen [114].

## 6. Bioactive Properties of *L. caerulea* and *A. melanocarpa* Fruits and Leaves

The present review gathers relevant studies on a variety of biological properties of *L. caerulea* L. (Figure 5a) and *A. melanocarpa* (Figure 5b) fruit and leaf extracts which exhibit antioxidant, anti-inflammatory, antitumoral, antibacterial and antifungal, anti-diabetic and anti-obesity effects, as well as hepatoprotective, cardioprotective, and neuroprotective potential (Figure 5).

### 6.1. Antioxidant Activity

Free radicals and reactive oxygen species are constantly generated in the human body, with high levels leading to oxidative damage. The role of antioxidant compounds is important, as free radicals tend to attack nucleic acids, lipids, and proteins, leading to inflammation, cancer, and a multitude of chronic diseases [42,115,116]. Polyphenols have the potential to suppress the formation of free radical precursors, but more prominently, they induce termination in lipid peroxidation chain reactions by donating an electron to the free radical, thus making it stable [116]. Among these compounds, flavonoids, flavonols, and phenolic acids exhibit antioxidant properties through the reducing character of phenolic –OH groups, but also through the property of chelating bi- and trivalent metals (Mg^2+^, Cu^2+^, Al^3+^, Fe^2+^, Zn^2+^). Flavonoids protect tissues against free radical oxygen species (ROS), especially superoxide radicals and membrane lipid peroxidation [18,113]. The structural elements necessary for the manifestation of protective action against ROS for flavonoids refers to the presence of the free OH group in position 3 of the benzopyran nucleus double bond between carbons 2–3; phenolic –OH groups on the phenyl from C-2 in “ortho”, respectively at C-3′ and C-4′; lipid solubility. The phenolic compounds have free radical scavenger properties, and inhibit lipid peroxidation much more obviously than the ligands themselves (protect the membranes of erythrocytes and platelets from the oxidative alteration induced by ROS, decrease the affinity of platelets towards vascular lesions, inhibit the oxidation of LDL, and are catalyzed by Cu^2+^).

Thanks to their high phenolic content, *L. caerulea* fruits are characterized by a notable antioxidant activity that usually ranges between 10,000–20,000 µmol Trolox equivalent/kg FW [81]. The antioxidants found in *L. caerulea* fruits take part in oxidative stress modulation mechanisms, their expression being determined by their bioactive compounds content and the extraction method [24,117].

It has been shown in an in vivo study that the antioxidants extracted from *L. caerulea* fruits led to decreased malondialdehyde (the final product of polyunsaturated fatty acid oxidation) content and an increase in superoxide dismutase and glutathione peroxidase (enzymes involved in oxidative damage protection) activity after radiation exposure, thus conferring irradiation protection [118].

In regard to *A. melanocarpa* fruit and leaf extracts, Cvetanović et al. (2018) conducted a study that aimed to compare their antioxidant activity using both DPPH and ABTS assays. The results showed that in the case of the DPPH assay, the highest antioxidant activity was exhibited by the leaf extract while in the case of the ABTS assay, the strongest scavenging activity was exerted by the fruit extract [87]. The differences between the two tests were most likely due to the fact that DPPH and ABTS assays count on different mechanisms of action.

Different types of *A. melanocarpa* leaf extracts were analyzed for antioxidant activity determination using DPPH, FRAP, ABTS, reducing power, inhibitory activity against lipid peroxidation [87], superoxide anion scavenging activity [94], chemiluminescence method (CL), luminol—H_2_O_2_ system, pH 8.9 [97], and β-carotene bleaching assay [89]. As expected, the antioxidant activity was correlated to a wide variety of biological activities, such as anti-inflammatory activity [95], antitumor activity [87], antimicrobial activity [87], and hypoglycemic and anti-neurodegenerative activity.

It was also reported in an in vivo study that an extract of *Aronia* leaf demonstrated antioxidant activity, accelerating skin epithelization when applied to the damaged skin of female rabbits [90].

Recent research established correlations between different methods of determining the antioxidant capacity through statistical evaluations in vitro—in vivo. Besides *Aronia* extracts, important constituents of flavones, flavonols and, in particular, phenolic acid classes are currently tested to evaluate their antioxidant capacity for possible use in the treatment of various conditions that involve the release of ROS [87,89,94,97].

### 6.2. Anti-Inflammatory Activity

The anti-inflammatory activity of *L. caerulea* fruits was demonstrated in vitro and in vivo. It was reported that the methanolic extract of Canadian haskap berries inhibited the production of TNF-α and IL-6 pro-inflammatory cytokines, as well as PGE2 and COX-2 inflammation enzymes in THP-1 derived macrophages stimulated by LPS. Considering the data available in the literature, the team took into consideration the possibility that the anti-inflammatory properties could be also involved in gene expression of the cytokines and enzymes [42]. In another in vitro study, *L. caerulea* L. fruits polyphenols increased the levels of Nrf2 and MnSOD antioxidant response proteins and inhibited TLR4-mediated inflammation in RAW264.7 cells [119].

In vivo, *L. caerulea* fruit extracts were found to exhibit anti-inflammatory activities against endotoxin-induced uveitis (an eye inflammation) in rats [27]. Another study evaluated the effect of oral administration of *L. caerulea* fruit polyphenols in a mouse paw edema model. The results showed that the treatment reduced serum levels of pro-inflammatory molecules, including MCP-1, IL-1β, IL-6, and TNF-α [119].

In mouse skeletal muscle tissue, *L. caerulea* extracts were found to lower fatigue-associated inflammation, increase energy storage, lower the accumulation of toxic metabolites, modulate apoptosis, and stimulate cell proliferation, thus enhancing muscle performance [120]. Another in vivo study in mice demonstrated that the iridoid-anthocyanin extract of *L. caerulea* fruits was responsible for anti-inflammatory activity during the intestinal phase of *Trichinella spiralis* infection [121]. Piekarska et al. (2022) reported that this immunotropic activity was most likely modulated by the extract through cytokines expression [121].

In contrast to many studies regarding the pharmacological activity of *L. caerulea* fruit extracts, little information is available regarding the biological activity of haskap leaves. Thus, it was reported that *L. caerulea* leaf extract has exerted anti-inflammatory effects by decreasing the secretion of pro-inflammatory molecules in LPS-stimulated RAW264.7 cells [122]. In this study, it was highlighted that the level of anti-inflammatory activity of the leaf extract was higher than the anti-inflammatory activity of the fruit extract [122]. A methanolic extract of haskap leaves was also more potent than the fruit extract, in that it stimulated the differentiation of stem cells into lymph nodal cells, modulating the immune system in response to *Streptococcus pyogenes* infection in a murine model [123].

As they contain many bioactive ingredients, *A. melanocarpa* fruits have possible use in the treatment of inflammatory diseases (Figure 5b). It has been shown that *Aronia* fruit extract ameliorates the severity of inflammation in mice with dextran sodium sulfate-induced ulcerative colitis, which is known as a model of inflammatory bowel diseases [124]. In another study, *Aronia* fruit extract had anti-inflammatory effects in an experimental model of inflammation caused by histamine and serotonin treatment [125]. Such extracts were also found to efficiently quench lipopolysaccharide-stimulated inflammation in mice cells and lower lipopolysaccharide-stimulated lipid peroxidation while showing no significant cytotoxicity [99]. *Aronia* fruit extract also inhibited the secretion of IL-6 in lipopolysaccharide-induced murine splenocyte [126] and blocked the expression of iNOS and COX-2 in rat macrophages [127]. Moreover, the extract of *A. melanocarpa* leaves had anti-inflammatory properties in LPS-stimulated RAW264.7 mouse macrophages by decreasing the expression of TNF-α and IL-6 pro-inflammatory cytokines [128].

### 6.3. Antitumor Activity

According to the literature, the most investigated biological activity of *A. melanocarpa* leaves is their anticancer activity. Different researchers reported the antitumoral activity of *A. melanocarpa* fruits. The rich phenolic content was also found to destroy certain types of tumoral cells, such as stem-like teratocarcinoma cells [129]. In another two studies, *A. melanocarpa* anthocyanin-rich extracts exerted antiproliferative effects in colon cancer cells, once again confirming its antitumor activity [130,131]. The studies have shown that the leaf extracts inhibited the growth of A-549 human lung cancer cells, LS-174T human colorectal cancer cells, HeLa human cervical cancer cells [87], HL60 human leukemic cells [128], and SK-Hep1 human hepatoma cells [132]. In [132], authors reported that the extract of *A. melanocarpa* leaves inhibited cell migration of SK-Hep1 human hepatoma cells.

Several researchers also reported the antitumor activity of *A. melanocarpa* fruits. *Aronia* fruits had radioprotective effects by increasing the survival rate in rats irradiated with γ rays and antimutagenic effects by lowering the damaging activities of some chemical mutagens [18]. These observations recommend the use of *Aronia* fruit extract as a suitable agent in cancer therapy.

### 6.4. Antibacterial and Antifungal Activity

The antimicrobial effect of *L. caerulea* fruits was observed against strains of *Bacillus subtilis*, *Kocuria rhizophila*, *Listeria monocytogenes*, *Escherichia coli*, *Lactobacillus acidophilus*, *Campylobacter jejuni* [133], *Staphylococcus aureus*, *Pseudomonas aeruginosa*, and *Salmonella enterica* [72]. Haskap extracts also inhibited the adhesion of some pathogenic microorganisms, such as *Enterococcus faecalis*, *Staphylococcus epidermidis*, *Streptococcus mutans*, and *Candida parapsilosis* [65].

In addition, *A. melanocarpa* leaves also possess antibacterial and antifungal activity. It was demonstrated that *A. melanocarpa* leaf extract caused morphological changes in the bacteria cells and inhibited DNA synthesis in meat spoilage and pathogenic bacteria [134]. Another study reported that *A. melanocarpa* leaf extract exerted antibacterial activity against *Proteus mirabilis*, *Proteus vulgaris*, *Bacillus subtilis*, *Staphylococcus aureus*, *Klebsiella pneumoniae*, and *Escherichia coli*, as well as antifungal activity against *Candida albicans* and *Aspergillus niger* [87]. The antibacterial effects were proven for *Aronia* fruit extracts, for instance against strains of *Bacillus cereus*, *Escherichia coli*, *Pseudomonas aeruginosa,* and *Staphylococcus aureus*. This antibacterial action was multiple, including biofilm formation inhibition and bacteriostasis [18]. It is believed that anthocyanins are responsible for the antibacterial activity of *A. melanocarpa* fruits. The scientific literature has shown that anthocyanins extracted from *A. melanocarpa* fruits exerted antibacterial activity by destroying the integrity of the cell wall and cell membrane of *E. coli*, leading to cell death [135].

*Aronia* fruit extracts also possessed antiviral properties, in particular against influenza viruses (including some drug-resistant strains) [129]. Myricetin and ellagic acid were the two main compounds in chokeberries responsible for the antiviral activity [129]. The antiviral bioactive compounds inhibited virus replication directly by blocking viral surface glycoproteins and indirectly by stimulating the immune system [136].

### 6.5. Anti-Diabetic and Anti-Obesity Activity

Many studies have indicated the beneficial effects of *L. caerulea* fruits in supporting the treatment of obesity and diabetes [83]. A Korean study coordinated by Chun et al. (2018) investigated the influence of *L. caerulea* fruit extract ingestion on mildly diabetic obese mice fed a high-fat diet. The results indicated significant weight loss and a decrease in the fat deposits in the abdominal wall and periovarian region, confirming the anti-obesity effects of haskap fruits [137]. Other studies confirmed that some fractions of *L. caerulea* fruit extracts had anti-diabetic, insulin-like effects, inhibiting starch decomposition in the intestine. They also directly stimulated insulin production [27].

Another study showed that the administration of an *L. caerulea* leaf infusion lowered the cholesterol level and inhibited the enzymes responsible for the absorption of simple sugars, which indicated the potential beneficial effects of haskap leaves in type 2 diabetes [83].

*Aronia* fruit extracts were also proven to have anti-diabetic activity. The treatment of diabetic rats with *A. melanocarpa* fruit juice reduced plasma glucose and triglyceride levels to the levels associated with non-diabetic rats [138]. Another study confirmed that *Aronia* fruits decreased blood glucose levels in diabetic mice, and showed a protective effect in rat RINm5F insulinoma cells, used as a cell culture model for pancreatic β-cells [139].

### 6.6. Hepatoprotective Activity

Another known biological property of haskap and *Aronia* fruits is liver protection. Thus, the treatment of *L. caerulea* with an ethanolic extract in an in vitro experimental model of free fatty acid-induced human HepG2 hepatocarcinoma cells was successful in ameliorating cellular steatosis through regulation of gene expression involved in lipid metabolism, such as sterol regulatory element-binding protein-1, CCAAR/enhancer-binding protein α, peroxisome proliferator-activated receptor gamma and fatty acid synthase and those of the fatty acid oxidation process [140]. The activity of insulin production stimulation through ROS inhibition and antioxidant mechanisms activation was demonstrated for cyanidin 3-O-glucoside, the main phenolic compound in haskap fruits [141]. The oral administration of *L. caerulea* fruit extract to mice with a high-fat diet caused a decrease in serum total cholesterol, triglycerides, and low-density lipoprotein levels and improved liver enzymes, suggesting hepatoprotective but also cardioprotective effects [111]. In a similar in vivo model, the administration of an enzymatic extract of haskap fruits for 84 days could significantly inhibit non-alcoholic fatty liver disease, demonstrating promising functional food properties [142]. This thorough study showed positive action on liver enzymes, catalase, superoxide dismutase, glutathione, and hepatic lipid peroxidation. The observed mechanisms of action involved hepatic glucokinase, phosphoenolpyruvate carboxykinase and glucose-6-phosphatase activity, expression of AMPKα1, AMPKα2, hepatic acetyl-CoA carboxylase 1 and adiponectin mRNAs, and synthesis of adipose tissue uncoupling protein 2. An enzymatic extract obtained by pectinase treatment of haskap solution also found significant inhibition of liver damage by pre-treatment with a dose of 200 mg/kg for 7 days in carbon tetrachloride-induced acute hepatic damage in mice, proving more favorable hepatoprotective effects than 100 mg/kg silymarin, an extract already used in commercial products [141]. This study demonstrated the capacity of haskap fruit extract to intervene in apoptotic pathways that could result in mitochondrial damage, cytochrome c release, and downstream activation of caspase-3, triggering the cleavage of cellular proteins such as PARP, cytokeratin 18, and other caspases. Moreover, the extract could stimulate the activation of hepatic antioxidant defense systems.

Likewise, an in vivo study conducted on rat models demonstrated that the rich-polyphenol extract of *A. melanocarpa* fruits showed beneficial effects on LPS-induced liver diseases. The mechanisms that assured the positive effects on liver damage are believed to be the reduction in bacterial displacement and improving gut microbiota imbalances, alongside releasing anti-inflammatory factors and activating anti-inflammatory pathways [143]. In another in vivo study conducted on rats exposed to cadmium, the anthocyanins extracted from *A. melanocarpa* fruits reduced the harmful effects caused by cadmium exposure. The extract administration caused the decrease in cadmium accumulation in the liver and kidneys, as well as the decrease in aspartate aminotransferase and alanine aminotransferase activity [144].

### 6.7. Other Biological Activities

In addition, haskap and *Aronia* fruit extracts have proven cardioprotective potential. Thus, the oral administration of *L. caerulea* fruit extract to obese mice improved liver enzymes, suggesting a hepatoprotective effect [137]. Moreover, the administration of *Aronia* fruit extract in a dose of 300 mg/day in a clinical study conducted on patients with metabolic syndrome during a period of two months indicated the ability to decrease systolic and diastolic blood pressure, total cholesterol, low-density lipoprotein cholesterol, and total triglyceride, proving it has a cardioprotective effect [145]. Besides the anti-hypertensive and cardioprotective effects of *Aronia* fruits demonstrated in this study, the treatment of aging rats with *Aronia* juice showed lower atherogenic risk and lower cardioprotective indices by lowering the proatherogenic low-density lipoprotein fraction [146].

Extracts of both species were revealed to possess neuroprotective activity. While few experiments on haskap extract have been conducted, some bioactive constituents, such as cyanidin 3-glucoside and other phenolic compounds, were shown to have neuroprotective effects. They lowered the probability of neurodegenerative disease and improved cognitive functions by enhancing neuronal growth and survival in adverse conditions (ethanol or glutamate-induced toxicity), lowering amyloid plaque formation, and strengthening synaptic responses [68]. A study presented by Meng et al. (2018) reported that *A. melanocarpa* anthocyanins might be effective in preventing Alzheimer’s disease. In an in vitro model of Alzheimer’s disease, the treatment of SH-SY5Y cells with *A. melanocarpa* anthocyanins inhibited Aβ1-42-induced apoptosis, decreased intracellular Ca^2+^ and ROS, and increased ATP and mitochondrial membrane potential [147]. The neuroprotective effects were demonstrated in another two studies by intraperitoneal injection of *Aronia* leaf extract, which exerted antioxidant activity in rat brain [148] and in vitro inhibition of cholinesterase activity of *Aronia* leaf extract, which might suggest that it could have beneficial effects in patients with Alzheimer’s disease [87].

In a recent study, the immunotropic activity of *L. caerulea* fruits was evidenced in *Trichinella spiralis*-infected mice. The authors indicate that fruit extract modulates the murine cellular immune response, most probably due to high levels of iridoids and anthocyanins [121].

## 7. Encapsulation and Delivery Systems of the Bioactive Compounds

The substances responsible for the biological activity of natural extracts are usually anthocyanins, flavonoids, some vitamins, and organic acid derivatives. They can be found in the plant extract, but in general for therapeutic use a higher concentration is needed, and therefore the plant extract is usually concentrated by solvent evaporation. The inherent instability is alleviated by encapsulation in a delivery system, where humidity is lowered, and oxidative stress from air- and light-promoted degradation processes are minimized [149,150,151]. There are multiple applications for the encapsulated natural components (Figure 6a). Such delivery systems can be applied in agriculture, food packaging, cosmetics, and other topical care products, and the administration of nutraceutical and pharmaceutical formulations. The principal role of the encapsulation system is to ensure the protection and controlled release of the bioactive substance. Such nanocarriers come in a variety of forms, such as nanocapsules, liposomes, solid lipid nanoparticles, polymeric nanoparticles, vesicles, mesoporous nanoparticles, microspheres, etc. (Figure 6b) [152,153,154]. For medical applications, the bioactive compounds interact with the body fluids, and usually many transformations occur. Therefore, using an encapsulation system can provide protection to the bioactive substances, achieving safe delivery. The ultimate goal is to provide a proper kinetic profile to the biologically active agents, protect them against various body fluids, and ensure a targeted delivery [155].

### 7.1. Encapsulation of A. melanocarpa or L. caerulea Bioactive Compounds for Oral Administration

Some of the components of a natural extract, such as those of *A. melanocarpa* or *L. caerulea*, can have a direct effect on various digestive infections, including colon cancer. The main components, polyphenols, flavonoids, and anthocyanins, are well studied for their antioxidant and antimicrobial activities or cytotoxicity. For oral administration of natural active substances, the need for a delivery system leads to specific requirements such as long-term controlled delivery avoiding the burst release effect, protecting the substances from low pH during stomach transit, delivery of substances at specific locations in the digestive system (see Figure 7), e.g., duodenum, jejunum, ileum, or colon [156]. The main flavonol in *Aronia* is quercetin, which has been extensively tested for its antitumoral applications, exhibiting also gastroprotective, antibacterial, and antiviral activities [157,158,159]. Quercetin suffers from low bioavailability when administered orally. Encapsulation of the quercetin and other bioactive compounds from *A. melanocarpa* or *L. caerulea*, such as caffeic, ferulic, gallic, or coumaric acids in the delivery system, protects these components inside the digestive tract and permits the release in the target region, fighting bacterial infections and tumor growth and aiding the healing process [160].

### 7.2. Encapsulation of L. caerulea and A. melanocarpa Extracts and Their Advantages

*A. melanocarpa* [162,163,164,165] and *L. caerulea* [24,82,121,166] are among the medicinal plants for which the literature reports multiple beneficial effects. Therefore, many bioactive compounds are extracted from these plants and are encapsulated into controlled release systems [167,168,169,170,171,172,173]. The literature reports various encapsulation systems for the active substances or natural extracts (Table 1).

Biodegradable polymers (such as polysaccharides, polyglycols, lactides, lipids, or proteins) and/or inorganic nanoparticles are used to obtain the delivery systems. The main advantage of these systems is that their morphology and composition can be adapted and tailored to the desired release profile to cross the biological barriers in the body.

The encapsulation of the natural extracts helps protect the active components (such as cyanidin-3-O-glucoside which is the predominant anthocyanin) from the digestive system [195]. After the release, some metabolites, such as gallic acid, ferulic acid, and 3,4-dihydroxybenzoic acid, are generated by the degradation of chlorogenic acid and anthocyanins from the natural extract. These extracts exhibit strong antioxidant, anti-cancer, anti-diabetic and anti-aging activities [25,29,72,117,120,196,197], being promising adjuvants in medicine.

In addition, the strong antioxidant and antimicrobial activities of the plant extract promotes them for other applications such as food packaging, protective cosmetics, or antimicrobial textiles [180,193,194].

In conclusion, the encapsulation of natural extracts in controlled release systems generates a series of advantages in the design of new drugs, but most studies have focused only on acute toxicity [198]. Long-term toxicity studies must be carried out in order to assess the safety and toxicity of a new system. Because various interactions may occur over time among the carrier, active substance, and biological environment, such studies are a mandatory requirement. At the cellular level, the impact of the nanomaterials used as carriers, being inorganic or biopolymer nanoparticles, can promote undesirable reactions that change the delicate balance between anti-inflammatory and inflammatory mechanisms [199]. Thus, in order to be officially approved as medicinal products, controlled release systems must be subjected to in vitro and in vivo tests on toxicology within the preclinical development phase.

## 8. Conclusions

*L. caerulaea* and *A. melanocarpa* fruits contain considerable amounts of polyphenolic compounds and anthocyanins compared with other commercial fruits. Such a rich phytochemical composition comes with associated health benefits for human consumption, with the literature indicating antitumoral and antimicrobial activities, and also neuroprotective, cardioprotective, and hepatoprotective actions. Due to their specific composition, anti-diabetic, anti-obesity, and anti-inflammatory activities were also reported.

The antioxidant effect is the most important characteristic of *L. caerulea* fruits, being owed to the phenolic content, besides vitamin C. It has been shown that *L. caerulea* presents an antioxidant activity three to five times higher in comparison with some more commonly consumed berries, such as blackberries or strawberries.

Anthocyanins represent approximately 25% of the polyphenol content found in *A. melanocarpa* fruits, derivatives of cyanidin being the most abundant. The main flavonol in *Aronia* is quercetin. Chlorogenic and neochlorogenic acids and tannins are also key constituents. All of these allow chokeberry to be considered one of the richest known sources of antioxidants, surpassing currants, cranberries, blueberries, elderberries, or gooseberries.

Many literature reports on *L. caerulaea* and *A. melanocarpa* fruits have emphasized that their biological properties were due to the synergistic activity of all phenolic compounds, instead of individual, specific substances. This opens new research opportunities in this field of study, due to the countless types of phenolic compounds found in fruits. A clear description of the cultivar conditions for the reported samples should be considered as a prospective research direction, as wide variation in the amount of bioactive phytochemicals has been reported.

Other parts of these species (leaves) were also found to be rich sources of bioactive compounds such as phenols, carotenoids, or chlorophylls. Loganic acid was the most abundant iridoid in haskap leaves while *Aronia* leaves have high levels of chlorogenic acid, caffeoylquinic acid derivatives, quercetin, and rutin. Unfortunately, the leaves are considered by many authors a byproduct or waste material, when they are in fact a valuable source of antioxidants and other compounds known to have healing, hypoglycemic, antineurodegenerative, or antimicrobial activity. Integrating the leaves as raw materials for these compounds will promote the development of a sustainable economy.

These vegetal species constitute a rich source of bioactive phytochemicals for the human diet, such as vitamins, tocopherols, folic acid, and carotenoids. Nevertheless, they are still not used at their full potential, as only small plantations are being industrialized. This review aimed to promote *L. caerulaea* and *A. melanocarpa* species as a healthy and valuable source of bioactive compounds, to increase their cultivation, processing, and commercial availability in the form of raw fruits or processed foods (jams, jellies, soft drinks, candied infusions or wine), as well as an important source of antioxidants as nutraceuticals for human health.

## Figures and Tables

**Figure 1 antioxidants-12-00951-f001:**
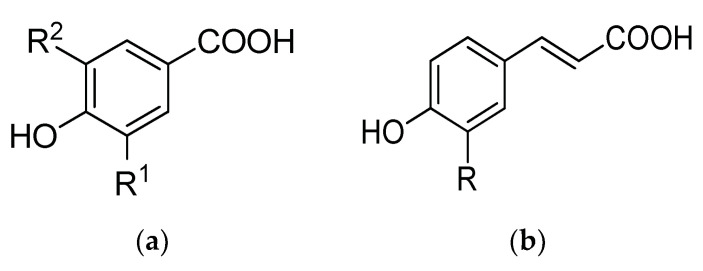
Chemical structure of (**a**) hydroxybenzoic acids (vanillic acid: R^1^ = OCH_3_, R^2^ = H; syringic acid: R^1^ = R^2^ = OCH_3_; gallic acid: R^1^ = R^2^ = OH); (**b**) hydroxycinnamic acids (caffeic acid: R = OH; ferulic acid: R = OCH_3_; p-coumaric acid: R = H) [52].

**Figure 2 antioxidants-12-00951-f002:**
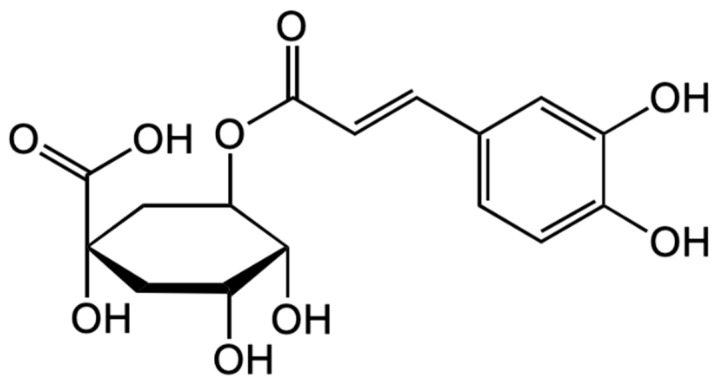
Chemical structure of chlorogenic acid (5-O-caffeoylquinic acid).

**Figure 3 antioxidants-12-00951-f003:**
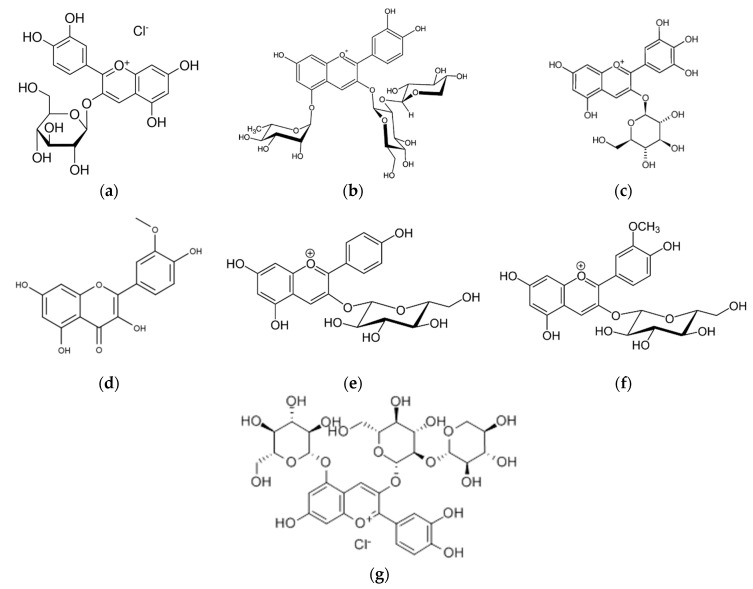
Chemical structures of the anthocyanins compounds: (**a**) cyanidin-3-glucoside, (**b**) cyanidin-3-rutinoside, (**c**) delphinidin-3-O-glucoside, (**d**) peonidin-3-O-rutinoIe, (**e**) pelargonidin-3-O-glucoside, (**f**) peonidin-3-O-glucoside, (**g**) cyanidin-3-sophorose-5-glucoside.

**Figure 4 antioxidants-12-00951-f004:**
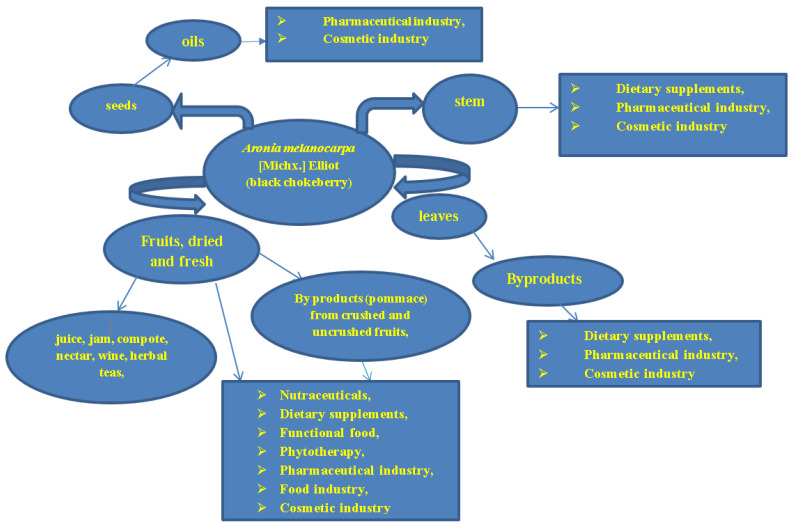
Proposed uses of *A. melanocarpa* L. and its byproducts.

**Figure 5 antioxidants-12-00951-f005:**
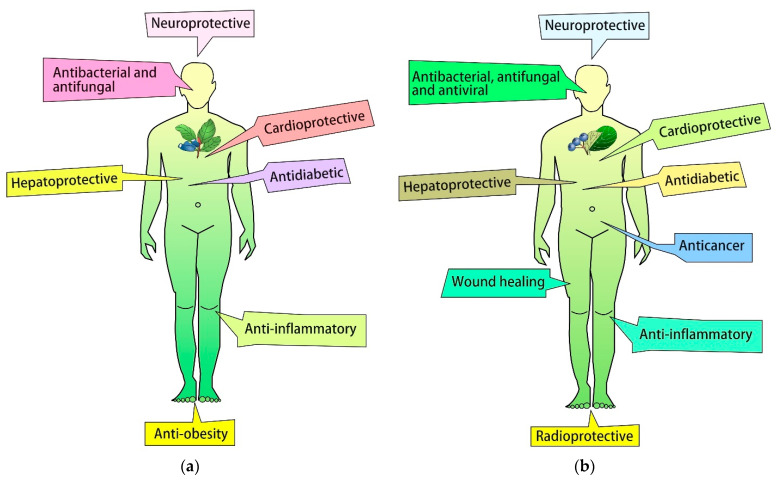
Pharmacological effects of *L. caerulea* (**a**) and *A. melanocarpa* (**b**).

**Figure 6 antioxidants-12-00951-f006:**
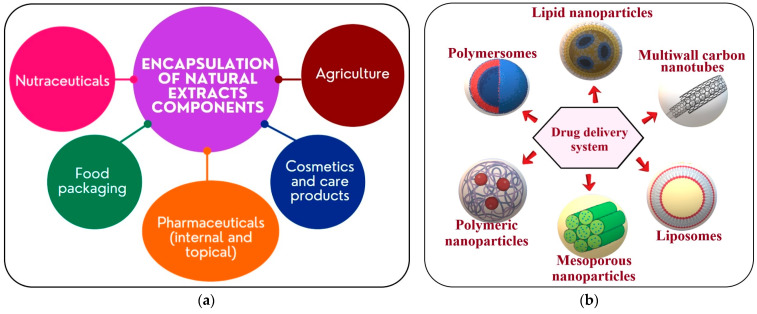
The encapsulation of natural extracts and their applications (**a**); Drug delivery systems for bioactive substances (**b**).

**Figure 7 antioxidants-12-00951-f007:**
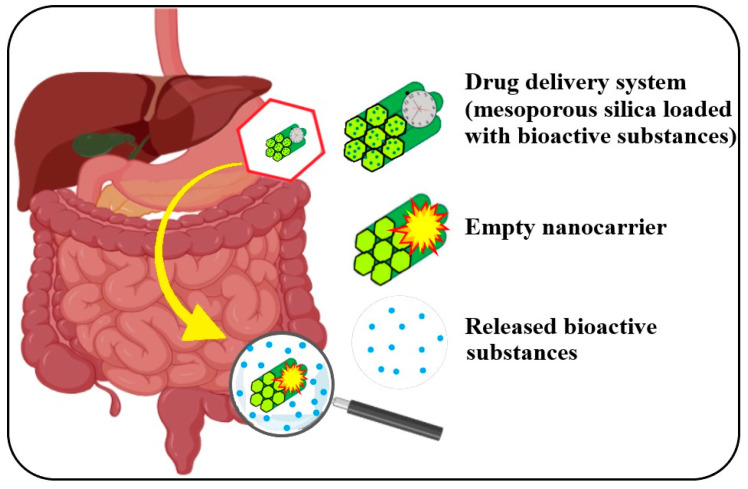
Digestive system disease treatment with mesoporous silica loaded with bioactive substances (realized with BioRender.com; accessed on 15th February 2023) [156,160,161].

**Table 1 antioxidants-12-00951-t001:** Controlled release systems with active substances available in *A. melanocarpa* and *L. caerulea*.

Encapsulation System	Tested Components	Source/Extract	Reference
Alginate particles	Anthocyanin rich extract	*L. caerulea*	[28,174]
Protein/lipid particles	Water extract	*L. caerulea*	[175]
Pea protein film	Leaf extracts	*L. caerulea*	[176]
Chitosan/starch	Anthocyanins	*L. caerulea*	[177]
SiO_2_	Anthocyanins	*L. caerulea*	[178]
Chitosan	Cyanidin-3-O-glucoside	*L. caerulea*	[179,180]
Alginate microparticles	Anthocyanins	*A. melanocarpa*	[168,181]
Chitosan nanocapsules	Anthocyanins	*A. melanocarpa*	[182,183]
Amylopectin nanoparticles	Anthocyanins	*A. melanocarpa*	[184,185]
MCM-41/ZnO	Polyphenols and flavonoids	*A. melanocarpa*	[186]
Maltodextrin microparticles	Polyphenols and anthocyanins	*A. melanocarpa*	[187,188,189]
Skimmed milk microparticles	Polyphenols and anthocyanins	*A. melanocarpa*	[187]
β-Cyclodextrin	Polyphenols	*A. melanocarpa*	[188]
Lecithin	Ethanol extract	*A. melanocarpa*	[190]
Starch extrudate	Polyphenols	*A. melanocarpa*	[191]
Polyvinyl alcohol/chitosan film	Polyphenols	*A. melanocarpa*	[192]
Polyurethane nanoweb	Polyphenols and flavonoids	*A. melanocarpa*	[193]
Polydimethylsiloxane	Anthocyanins	*A. melanocarpa*	[194]

## Data Availability

The data is contained within the article.

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
