# Peer review of "Health Benefits of Antioxidant Bioactive Compounds in the Fruits and Leaves of Lonicera caerulea L. and Aronia melanocarpa (Michx.) Elliot"

_antioxidants, 2023, doi:10.3390/antiox12040951_

Round 1
Reviewer 1 Report
Dear Editor and Authors,
The manuscript ‘Health benefits of antioxidant bioactive compounds from Lonicera caerulea L. and Aronia melanocarpa (Michx.) Elliot’ by Bogdan-Stefan Negreanu-Pirjol, Ovidiu Cristian Oprea, Ticuta Negreanu-Pirjol, Florentina Nicoleta Roncea, Ana-Maria Prelipcean, Oana Craciunescu, Victoria Artem, Aurora Ranca, Ludmila Motelica , Anca Cristina Lepadatu, Madalina Cosma, Dan Razvan Popoviciu is a review on : Lonicera caerulaea L. and Aronia melanocarpa (Michx.) Elliot phytochemical composition and health-beneficial properties as well as methods of extracts encapsulation and drug delivery. The Authors collected a decent number of references (232.) on the topic.
The manuscript is interesting and worth publication in Antioxidants.
I have found some minors that after improvement should increase the quality of the manuscript:
Line 138 Delete ‘from’ and rewrite the sentence. I suggest: The honeysuckle (honey berry) fruits ripen early…
Line 144 and further
Latin names should be in italics
Line 165 ‘Aronia fruits (black currants)’
Blackcurrants are Ribes nigrum, it may be confusing
Line 169
In my opinion the main reason aronia is grown at an industrial scale is that the fruits are full in stable colorants, process easily into jams and by the way they are healthy. I would not state that commercial reason to grown the plants is that there are beneficial to human health. Give also other reasons.
Yours sincerely,
Author Response
The manuscript ‘Health benefits of antioxidant bioactive compounds from Lonicera caerulea L. and Aronia melanocarpa (Michx.) Elliot’ by Bogdan-Stefan Negreanu-Pirjol, Ovidiu Cristian Oprea, Ticuta Negreanu-Pirjol, Florentina Nicoleta Roncea, Ana-Maria Prelipcean, Oana Craciunescu, Victoria Artem, Aurora Ranca, Ludmila Motelica , Anca Cristina Lepadatu, Madalina Cosma, Dan Razvan Popoviciu is a review on : Lonicera caerulaea L. and Aronia melanocarpa (Michx.) Elliot phytochemical composition and health-beneficial properties as well as methods of extracts encapsulation and drug delivery. The Authors collected a decent number of references (232.) on the topic.
The manuscript is interesting and worth publication in Antioxidants. I have found some minors that after improvement should increase the quality of the manuscript:
R: Thank you for your dedicated time for revising this review. We provide below the revised text, taking into account your valuable observations. All revisions were marked with yellow in the manuscript.
- Line 138 Delete ‘from’ and rewrite the sentence. I suggest: The honeysuckle (honey berry) fruits ripen early…
The sentence was corrected: “The honeysuckle (honey berry) fruits ripen early…”
- Line 144 and further Latin names should be in italics
All the manuscript was revised and Latin names were provided in italics.
- Line 165 ‘Aronia fruits (black currants)’. Blackcurrants are Ribes nigrum, it may be confusing.
The sentence was corrected as “Aronia fruits (Aronia melanocarpa)”.
- Line 169 – In my opinion the main reason aronia is grown at an industrial scale is that the fruits are full in stable colorants, process easily into jams and by the way they are healthy. I would not state that commercial reason to grown the plants is that there are beneficial to human health. Give also other reasons.
The phrase was revised to include valuable reasons for Aronia valorization. It was reformulated as:
“Besides their valuable composition, rich in bioactive compounds with important biological properties, these fruits could be valorized, at an industrial scale, by easy processing into edible jams and processing as stable and natural food colorants [31].”
Reviewer 2 Report
In this review, the authors have attempted to provide information on the bioactive compounds in the fruits and leaves of Lonicera caerulea L. and Aronia melanocarpa (Michx.) Elliot. Such review could be especially relevant today when the interest in natural remedies has increased. So, for the topic of this publication, the selected objects are very important. In assessing the overall quality of the article, the first thing to note is the general inconsistency, especially in the use of common L. caerulea names. The cited references were not evaluated critically, so there are a number of professional errors. Throughout the manuscript, it is necessary to clarify whether the quotations correspond to references, so that the reader must receive accurate and correct information.
Title. The title of the article needs to be clarified, because it is not clear what parts of the specified plant species will be discussed, i.e. leaves, flowers, fruits or others.
Abstract. Only the value of fruits of L. caerulea and A. melanocarpa was summarized. However, the text indicates the provided information and about the leaves (p. 5, line 179, p. 8, line 313). So, the abstract must be more detailed.
Keywords. I would suggest not repeating words that are already in the title.
Introduction. I propose this section for major revision. It provides general information about causes of antioxidant stress and how the antioxidant mechanism works in aerobic organisms (p. 2, lines 68) etc. In my opinion, 2-7 references should be abandoned as well.
The authors present incorrect information in the botanical sense. L. macckii, L. morrowii, L. fragrantissima and L. japonica are not "varieties of L. caerulea", but different species. Fruits of these species have the potential to be toxic and are considered poisonous to humans. Perhaps the fruits of these Lonicera species can be applied in medicine, but this must be considered from another point of view. I suggest deleting Fig. 1.
I would suggest the authors to explore the information provided by World Flora Online. The name of L. kamtschatica Pojark. is a synonym of L. caerulea.
http://www.worldfloraonline.org/taxon/wfo-0000359581
I also recommend to study the article presenting investigations of Lonicera caerulea subspecies: Plekhanova, M.N. Blue honeysuckle (Lonicera caerulea L.) – a new commercial berry crop for temperate climate: Genetic resources and breeding. Acta Hortic. 2000, 538, 159–164. Genetic diversity was not evaluated in the review, but it could help the authors to avoid such serious mistakes.
Section 2. Lonicera caerulea; fruits or berries? I would recommend to use one term - fruits.
The authors make an incorrect statement on p.5, line 185-186. The fruits of L. caerulea certainly do not accumulate the highest ascorbic acid content. On the other hand, does this cited reference [37] really present data on ascorbic acid levels?
37. Rahman, S.; Mathew, S.; Nair, P.; Ramadan, W.S.; Vazhappilly, C.G. Health benefits of cyanidin-3-glucoside as a potent modulator of nrf2-mediated oxidative stress. Inflammopharmacology 2021, 29, 907-923.
And the authors of the cited source [73] provide data that fruit of L. caerulea accumulate of 18.6 mg/100g FW of ascorbic acid on average. As Jurikova and Matuškovič (2007) reported, the mean value of ascorbic acid was 55.716 mg
per 100 g in fruits of L. caerulea. These data are already inconsistent with the statements of the manuscript.
I also did not find any information about leaves of L. caerulea in this section, although the title of the section mentions the leaves (p. 5, line 179).
Section 3. Please correct this reference in the list (cited in p. 9, line 346) as follows:
21. Shahin, Lubana; Sheriff S. Phaal; Brajesh N. Vaidya; James E. Brown; and Nirmal Joshee. 2019. "Aronia (Chokeberry): an underutilized, highly nutraceutical plant." Journal of Medicinally Active Plants 8:46-63.
Section 5. Again, it is necessary to review the references cited in this section. The authors here too extensively examine the encapsulation and delivery systems of various biologically active compounds of other plants. I would suggest the authors to critically analyze this section and make sure that the cited references are really essential. In my opinion, this section should discuss the preparation and absorption of various forms of biologically active compounds that have been identified in L. caerulea and A. melanocarpa.
Conclusions: Studies on fruits were summarized, but bioactive compounds found in the leaves of these plants are not mentioned. And the same misleading statement about ascorbic acid was repeated again (p. 19, line 752).
Small remarks:
1. In the list of references, it is necessary to correct the Latin names of plants because they do not meet the requirements applied in botany: The name of the plant genus is always capitalized even in the middle of the sentence: Lonicera, Aronia.
2. When the name of the plant species is repeated in the text, the genus should be written in abbreviated form - L. caerulea. In this respect, the authors do not follow the rule throughout the manuscript.
3. When the Latin name of the plant is mentioned in the text for the first time, the author's initials should be also indicated, e.g. Lonicera caerulea L. Later, the initials of authors are no longer written.
Author Response
In this review, the authors have attempted to provide information on the bioactive compounds in the fruits and leaves of Lonicera caerulea L. and Aronia melanocarpa (Michx.) Elliot. Such review could be especially relevant today when the interest in natural remedies has increased. So, for the topic of this publication, the selected objects are very important. In assessing the overall quality of the article, the first thing to note is the general inconsistency, especially in the use of common L. caerulea names. The cited references were not evaluated critically, so there are a number of professional errors. Throughout the manuscript, it is necessary to clarify whether the quotations correspond to references, so that the reader must receive accurate and correct information.
Thank you for your dedicated time for revising this review. We provide below the revised text, taking into account your valuable observations. All revisions were marked with yellow in the manuscript.
- Title. The title of the article needs to be clarified, because it is not clear what parts of the specified plant species will be discussed, i.e. leaves, flowers, fruits or others.
The title was accordingly revised as: “Health Benefits of Antioxidant Bioactive Compounds in the Fruits and Leaves of Lonicera caerulea L. and Aronia melanocarpa (Michx.) Elliot”
- Abstract: Only the value of fruits of L. caerulea and A. melanocarpa was summarized. However, the text indicates the provided information and about the leaves (p. 5, line 179, p. 8, line 313). So, the abstract must be more detailed.
We are thankful for pointing out this inconsistency. We have included relevant information in Abstract section: “Also, non edible leaves of genus Aronia became more extensively analyzed as a byproduct or waste material for its high content, mainly in polyphenols, flavonoids, phenolic acids and small amounts of anthocyanins, as ingredients in nutraceuticals, tea herbals, bio-cosmetics, cosmeceuticals, food and pharmaceutical industry”
- Keywords: I would suggest not repeating words that are already in the title.
The Keywords were revised accordingly as: “haskap, aronia, polyphenols, anthocyanins, flavonoids, iridoids, flavonoids, antioxidant, anti-inflammatory, antimicrobial, antitumor, hepatoprotective, encapsulation”.
- Introduction: I propose this section for major revision. It provides general information about causes of antioxidant stress and how the antioxidant mechanism works in aerobic organisms (p. 2, lines 68) etc. In my opinion, 2-7 references should be abandoned as well.
We agree with the esteem reviewer, the introduction section was too general and educational. We have condensed the section and removed the less important paragraphs.
- The authors present incorrect information in the botanical sense. macckii, L. morrowii, L. fragrantissimaand L. japonica are not "varieties of L. caerulea", but different species. Fruits of these species have the potential to be toxic and are considered poisonous to humans. Perhaps the fruits of these Lonicera species can be applied in medicine, but this must be considered from another point of view. I suggest deleting Fig. 1.
We are grateful to the esteem reviewer for pointing out this mistake. You're right, these are different invasive species. Regarding different varieties, we could refer to Aurora, Berry Blue, Indigo Gem, Wojtek, Blizzard, Tundra, Happy Giant, Beauty, Strezwczanka, Jugana, Vostorg, Blue Banana and many others depending of the grower’s location or territory and intended use of the berries. Also, Figure 1 has been deleted
- I would suggest the authors to explore the information provided by World Flora Online. The name of kamtschaticaPojark. is a synonym of L. caerulea.
Thank you for your suggestions. We have used across the manuscript only Lonicera caerulaea name.
- I also recommend to study the article presenting investigations of Lonicera caerulea subspecies: Plekhanova, M.N. Blue honeysuckle (Lonicera caerulea L.) – a new commercial berry crop for temperate climate: Genetic resources and breeding. Acta Hortic. 2000, 538, 159–164. Genetic diversity was not evaluated in the review, but it could help the authors to avoid such serious mistakes.
We are thankful for this valuable suggestion.
- Section 2. Lonicera caerulea; fruits or berries? I would recommend to use one term - fruits.
The sentences containing fruits and berries were revised accordingly.
- The authors make an incorrect statement on p.5, line 185-186. The fruits of L. caerulea certainly do not accumulate the highest ascorbic acid content. On the other hand, does this cited reference [37] really present data on ascorbic acid levels?
- Rahman, S.; Mathew, S.; Nair, P.; Ramadan, W.S.; Vazhappilly, C.G. Health benefits of cyanidin-3-glucoside as a potent modulator of nrf2-mediated oxidative stress. Inflammopharmacology 2021, 29, 907-923.
This reference was replaced with a recent one. We are thankful for pointing out this mistake. Was the highest level among various cultivars. We have rephrased the sentence accordingly.
Gorzelany, J.; Basara, O.; Kapusta, I.; Paweł, K.; Belcar, J. Evaluation of the Chemical Composition of Selected Varieties of L. caerulea var. kamtschatica and L. caerulea var. emphyllocalyx. Molecules 2023, 28, 2525.
- And the authors of the cited source [73] provide data that fruit of caeruleaaccumulate of 18.6 mg/100g FW of ascorbic acid on average. As Jurikova and Matuškovič (2007) reported, the mean value of ascorbic acid was 55.716 mg
per 100 g in fruits of L. caerulea. These data are already inconsistent with the statements of the manuscript.
We rephrased the sentence. In the latest reference [33] the average value is 53.5 mg/100 g which is close to the one reported in Jurikova and Matuškovič (2007). Nevertheless, the reported literature values are always related to many variables, that authors should report.
- I also did not find any information about leaves of caeruleain this section, although the title of the section mentions the leaves (p. 5, line 179).
We have enriched the manuscript with relevant information for both species in section 4, 5 and 6:
Chemical composition of A. melanocarpa L. leaves differs from cultivar, to climatic zones and maturity stage, extraction method used as mentioned in numerous articles [18, 74-76]. Mixed spectrophotometric and chromatographic methods determined the chemical composition of Aronia leaves, from total phenolic content expressed as flavonols, flavonoids, anthocyanins, phenolic acids [74, 76-82], small amounts of chlorophyll, carotenoids [81], carbohydrates, macro and microelements, fibers, and crude fat [83], with most of high levels for chlorogenic acid, caffeoylquinic acid derivatives, quercetin, rutin, sorbitol, small amounts of anthocyanins, chlorophyll, carotenoids, macro and microelements, fibers identified mostly in young leaves compared to mature and old leaves (Figure 4). The extracts obtained from nonedible A. melanocarpa L. leaves are a valuable source as potential use as antioxidant, healing, hypoglycemic, antineurodegenerative, moderate antimicrobial activity. Although is a valuable raw material for developing a sustainable economy there is still little information about A. melanocarpa L, regarding leaves, considered by some authors, a byproduct or a waste material [76].
Figure 4. A. melanocarpa L.and its byproducts proposed uses
........................................................
4.2.4. Bioactive Compounds of Lonicera caerulea Leaves
The leaves of haskap were recently found as an important source of bioactive compounds. According to the variety, the total phenolic content varied between 32.127 and 52.399 mg/g, carotenoids content was between 1.848-2.876 mg/g, while chlorophyll A and chlorophyll B were present in concentrations of 10.642-14.003 and 4.463-5.575 mg/g, before storage [74]. The leaf extract prepared by ultrasonication in distilled water, at 30 °C, for 60 min contained several phenolic acids, such as loganic acid with anti-inflammatory properties acting against hyaluronidase and hypolipidemic potential acting against lipase, chlorogenic acid with antidiabetic potential acting against α-glucosidase, and caffeic acid with antioxidant and anti-inflammatory activity [74]. An extract prepared by aqueous two-phase flotation using ammonium sulfate-ethanol presented increased yields of flavonoids, such as rutin, luteolin-7-O-glucoside and diosmin, which provided increased antioxidant capacity [75,76]. Vitamin C was also determined in the aqueous extract of haskap leaves prepared at the Soxhlet equipment operated at 105 °C, for 20 h, but at a lower value (5.1 mg/100 g) than in berries extract (6.9 mg/100 g) and polysaccharides (7%), together with flavones, flavanones and tanning compounds, which could provide antibacterial and antiviral activity [77].
........................................
In contrast to many studies regarding the pharmacological activity of L. caerulea fruit extracts, little information is available regarding the biological activity of haskap leaves. Thus, it was reported that L. caerulea leaves extract has exerted anti-inflammatory effects by decreasing the secretion of pro-inflammatory molecules in LPS‑stimulated RAW264.7 cells [114]. In this study, it was highlighted that the level of anti-inflammatory activity of the leaves extract was higher than the anti-inflammatory activity of the fruits extract [113]. A methanolic extract of haskap leaves was also more potent than the fruit extract, in that it stimulated the differentiation of stem cells into lymph nodal cells, modulating the immune system in response to Streptococcus pyogenes infection in a murine model [114].
12) Section 3. Please correct this reference in the list (cited in p. 9, line 346) as follows:
- Shahin, Lubana; Sheriff S. Phaal; Brajesh N. Vaidya; James E. Brown; and Nirmal Joshee. 2019. "Aronia (Chokeberry): an underutilized, highly nutraceutical plant." Journal of Medicinally Active Plants 8:46-63.
The reference was accordingly revised as “Shahin, L.; Phaal, S.S.; Vaidya, B.N.; Brown, J.E.; Joshee, N. Aronia (Chokeberry): an underutilized, highly nutraceutical plant. Journal of Medicinally Active Plants 2019, 8, 46-63.”
- Section 5. Again, it is necessary to review the references cited in this section. The authors here too extensively examine the encapsulation and delivery systems of various biologically active compounds of other plants. I would suggest the authors to critically analyze this section and make sure that the cited references are really essential. In my opinion, this section should discuss the preparation and absorption of various forms of biologically active compounds that have been identified in caeruleaand A. melanocarpa.
We have condensed the section dedicated to the encapsulation of active compounds, eliminating more than half of the references. We have followed the valuable advice and limited the examples to the active compounds that have been identified in L. caerulea and A. melanocarpa and their extracts.
- Conclusions:Studies on fruits were summarized, but bioactive compounds found in the leaves of these plants are not mentioned. And the same misleading statement about ascorbic acid was repeated again (p. 19, line 752).
The conclusions section was remade with paragraphs dedicated to the bioactive compounds found in leaves. The statement about vitamin C was removed.
Other parts of these species (leaves) were also found to be a rich source of bioactive compounds like phenols, carotenoids or chlorophylls. Loganic acid was the most abundant iridoid in haskap leaves while Aronia leaves have high levels of chlorogenic acid, caffeoylquinic acid derivatives, quercetin and rutin. Unfortunately, the leaves are considered by many authors a byproduct or waste material, while they are in fact a valuable source of antioxidants and other compounds known to have healing, hypoglycemic, antineurodegenerative or antimicrobial activity. Integrating the leaves as raw materials for these compounds will promote developing a sustainable economy.
15) Small remarks:
- In the list of references, it is necessary to correct the Latin names of plants because they do not meet the requirements applied in botany: The name of the plant genus is always capitalized even in the middle of the sentence: Lonicera, Aronia.
We are grateful for this observation. The References were revised to correct the Latin names of plants. Unfortunately, as the references are introduced by EndNote, they are reverted to normal text automatically. This correction will be done again in proofing phase when the code in Visual Basic that accompanies the references is removed and changes become persistent.
- When the name of the plant species is repeated in the text, the genus should be written in abbreviated form - L. caerulea. In this respect, the authors do not follow the rule throughout the manuscript.
The manuscript was revised accordingly using the abbreviated name.
- When the Latin name of the plant is mentioned in the text for the first time, the author's initials should be also indicated, e.g. Lonicera caeruleaL. Later, the initials of authors are no longer written.
The manuscript was revised accordingly using only L. caerulea.
Reviewer 3 Report
The article ‘Health benefits of antioxidant bioactive compounds from Lonicera caerulea L. and Aronia melanocarpa (Michx.) Elliot’, presented for review is very interesting. However, some issues need to be clarified or supplemented. The comments are included below.
Title
The title is worded correctly and accurately reflects the content.
Abstract
- I consider repeating Aronia melanocarpa twice unnecessary.
Other:
- In the case of the Lonicera caerulea L, new fruit varieties with a rounder shape and higher extract have appeared in recent years. If the authors have access to this information, please complete it.
- Maybe it's worth paying more attention to Aronia melanocarpa (Michx.) Elliot seed oil. In recent years, there have been a number of publications in this field. For example, publications by Piasecka and co-authors.
- It is worth mentioning that black chokeberry is a valuable raw material for the production of mixed preserves. The combination of aronia with other raw materials has a positive effect on their health-promoting value and has a positive effect on the taste.
- In recent years, Lonicera caerulea L and Aronia melanocarpa have been used for the production of creamy honey. According to the reviewer, this is worth emphasizing.
- Line 248 - The percentage of cyanidin 3-glucoside may be much higher. Such information is included, for example, in publication 113 cited in the paper: (Grobelna, A.; Kalisz, S.; Kieliszek, M.; Giurgiulescu, L. Blue honeysuckle berry (lonicera caerulea l.), as raw material, is particularly predisposed to the production of functional foods. Carpathian J Food Sc 2020, 12, 144-155.)
- Why the authors omitted the issue of vitamin C content in haskap berry. According to the literature, Lonicera caerulea L can contain up to 180 mg of vitamin C in 100 g of fresh weight.
- In the case of Lonicera caerulea L, the content of iridoids is important. Please expand on this.
Author Response
The article ‘Health benefits of antioxidant bioactive compounds from Lonicera caerulea L. and Aronia melanocarpa (Michx.) Elliot’, presented for review is very interesting. However, some issues need to be clarified or supplemented. The comments are included below.
Thank you for your dedicated time for revising this review. We provide below the revised text, taking into account your valuable observations. All revisions were marked with yellow in the manuscript.
- The title is worded correctly and accurately reflects the content.
The title was revised to include fruits and leaves of each species.
- I consider repeating Aronia melanocarpa twice unnecessary.
The abstract was corrected. Other information have been also added.
- Other: - In the case of the Lonicera caerulea L, new fruit varieties with a rounder shape and higher extract have appeared in recent years. If the authors have access to this information, please complete it.
Unfortunately, we did not manage to identify literature for these varieties.
- Maybe it's worth paying more attention to Aronia melanocarpa (Michx.) Elliot seed oil. In recent years, there have been a number of publications in this field. For example, publications by Piasecka and co-authors.
We are thankful to the esteem reviewer for pointing this valuable resource. We have updated the section 3:
Recently seeds of A. melanocarpa seeds were used as raw materials for extraction of high-quality oil by Piasecka et al. [34,35]. Chokeberry oil was found to have the highest linoleic acid content among all studied oils (raspberry, blackberry and chokeberry), while the major tocols fraction was α-tocopherol [36].
- It is worth mentioning that black chokeberry is a valuable raw material for the production of mixed preserves. The combination of aronia with other raw materials has a positive effect on their health-promoting value and has a positive effect on the taste.
We have introduced the following information at the reviewer valuable suggestion:
Some recent research indicates the possible use of chokeberry pomace extract as an additive for apple juice, helping with its preserve and positively affecting the color and taste due to its antioxidant colorants [35,37,38]. The high anthocyanin content made chokeberry pomace suitable as an ingredient for chitosan-based packaging films with capacity to indicate pH variations [39].
- In recent years, Lonicera caerulea L and Aronia melanocarpa have been used for the production of creamy honey. According to the reviewer, this is worth emphasizing.
We have introduced the following information at the reviewer valuable suggestion:
At the same time, by enriching rape honey with chokeberry fruit additive the antioxidant activity increased 3 to 15 times, and intensifies its antibacterial and antiviral activities [40]. Similar strong activities were also reported by [41], which recommends the use of chokeberry and honeysuckle ethanol infusion as a food additive, especially against Listeria monocytogenes.
- Line 248 - The percentage of cyanidin 3-glucoside may be much higher. Such information is included, for example, in publication 113 cited in the paper: (Grobelna, A.; Kalisz, S.; Kieliszek, M.; Giurgiulescu, L. Blue honeysuckle berry (lonicera caerulea l.), as raw material, is particularly predisposed to the production of functional foods. Carpathian J Food Sc 2020, 12, 144-155.)
Thank you for pointing out this inconsistency. We have corrected the value with the one reported in the suggested study.
The most abundant is cyanidin-3-glucoside, representing around 79-92% [65].
- Why the authors omitted the issue of vitamin C content in haskap berry. According to the literature, Lonicera caerulea L can contain up to 180 mg of vitamin C in 100 g of fresh weight.
We have updated the information with:
“The ascorbic acid content varies between 30.5 to 186.6 mg·100 g−1, with an average of 53.5 mg·100 g−1, this fruit being regarded as a “superfood” [33].”
- In the case of Lonicera caerulea L, the content of iridoids is important. Please expand on this.
We have expanded the section for iridoids:
“Iridoids (a group of monoterpenoids) are one of the main bioactive compound classes (especially antitumoral, antibiotic, hepato- and neuroprotective, hypotensive, anti-inflammatory) found in haskap berries. This is important since few fruits are known to contain significant amounts. Loganic acid is dominant in haskap, while other iridoids, such as loganin and loganin pentosides, secologanin, secoxyloganin, sweroside, pentosyl-sweroside, are found in lower concentrations”
.........................................
In a recent study, the immunotropic activity of L. caerulea fruits was evidenced in Trichinella spiralis infected mice. The authors indicate that fruit extract modulates murine cellular immune response most probably due to high levels of iridoids and anthocyanins [120].
Round 2
Reviewer 2 Report
Authors made corrections taking into account suggestions and comments.